# How are public engagement health festivals evaluated? A systematic review with narrative synthesis

**Susannah Martin[1], Charlotte Chamberlain[1], Alison Rivett[2], Lucy E. Selman[1]***

**1** Palliative and End of Life Care Research Group, Population Health Sciences, Bristol Medical School, University of Bristol, Bristol, United Kingdom, **2** Public Engagement Team, University of Bristol, Bristol, United Kingdom

* lucy.selman@bristol.ac.uk

**Data Availability Statement:** All data is available from the referenced study reports included in the review.

## Abstract

The evaluation of public engagement health festivals is of growing importance, but there has been no synthesis of its practice to date. We conducted a systematic review of evidence from the evaluation of health-related public engagement festivals published since 2000 to inform future evaluation. Primary study quality was assessed using the Mixed Methods Appraisal Tool. Extracted data were integrated using narrative synthesis, with evaluation methods compared with the Queen Mary University of London public engagement evaluation toolkit. 407 database records were screened; eight studies of varied methodological quality met the inclusion criteria. Evaluations frequently used questionnaires to collect mixed-methods data. Higher quality studies had specific evaluation aims, used a wider variety of evaluation methods and had independent evaluation teams. Evaluation sample profiles were often gender-biased and not ethnically representative. Patient involvement in event delivery supported learning and engagement. These findings and recommendations can help improve future evaluations. (Research Registry ID reviewregistry1021).

## Introduction

Engagement and collaboration with the public are increasingly recognised as a core aspect of all research and particularly in health-related research [1, 2]. Reasons for such engagement include conversing with the public about research to raise awareness and trust; conducting citizen science; using two-way dialogue to inform and improve research; disseminating research results and sharing knowledge; and influencing policy [3, 4].

There are many overlaps between public engagement (PE) and the long-standing practice of Patient & Public Involvement (PPI) in medical and healthcare research. Commonly accepted definitions of these terms are given below, as set out by leading organisations in these two fields, the National Coordinating Centre for Public Engagement (NCCPE) and the National Institute for Health Research (NIHR) respectively.

**Funding:** LES received research and salary funding from the Wellcome Trust for this work (grant 218780/Z/19/Z). The funders had no role in study design, data collection and analysis, decision to publish, or preparation of the manuscript.

**Competing interests:** The authors have declared that no competing interests exist.

## Public engagement

The myriad of ways in which the activity and benefits of higher education and research can be shared with the public. Engagement is by definition a two-way process, involving interaction and listening, with the goal of generating mutual benefit [5].

## Patient & Public Involvement

Research being carried out 'with' or 'by' members of the public rather than 'to', 'about' or 'for' them. It is an active partnership between patients, carers and members of the public with researchers that influences and shapes research [6].

The definitions above demonstrate that whilst PPI is a relatively tightly defined concept as understood by healthcare practitioners and researchers, PE is a much more amorphous term [7, 8], encompassing many ways of engaging with the public and not necessarily just about research. PE, particularly when it involves engagement with specific research projects, or research-related matters (e.g. research ethics), rather than engagement around a wider subject area or topic, is sometimes specifically referred to as 'Public Engagement with Research' (PER). This part of the engagement spectrum is where there are most overlaps with PPI. Whereas PPI is generally a formally defined process within a healthcare research project, PE activities are often more informal, sometimes ad-hoc and can be delivered in a multitude of ways for a wide variety of audiences [9, 10].

If the precise meaning of engagement is vague, then a catch-all definition of 'the public' is even harder to pin down [11, 12]. What is commonly accepted in the PE sphere is that 'the public' should never be considered as one single entity, but a multi-dimensional spectrum of people with widely varying levels of expertise, lived experiences, interests, opinions and so on [13, 14]. It is critical that any PE activity is tailored to the specific audience it is aimed at, perhaps even co-developed with that group of people. In the context of this paper the understanding of 'publics' as *"gatherings of people, things, objects and ideas convened around a matter of concern."* as derived by Facer (2020) is helpful [15].

PPI and PE both play an important role in research related to human health. The UK's National Health Service (NHS) and the USA's Institute of Medicine support the co-production of healthcare plans with patients, increasing patient control over their health and emphasising disease prevention [16, 17]. People may therefore, more than ever, have reason to seek out and engage with health-related research. While relatively few members of the public have the opportunity to take part in a formal PPI process, public engagement opportunities might more readily present themselves.

Science festivals are one increasingly popular format for communication of, and public engagement with, health research [18]. Such festivals offer audiences a time-limited opportunity to engage directly with scientists and research [18, 19], but vary in their budget, venues, activity format, size and theme. With the proliferation of PE activity comes the need to understand how specific types of PE such as festivals work, who they work for and why [18]. Good quality evaluation of science and health-related festivals, with reflection and learning from current evaluation practice, is therefore essential [19–21]. A previous review of science festival evaluation by Peterman and colleagues [21] examined the methods and results reported in published science festival evaluations and research. Their review examined the literature from an expert standpoint within the context of visitor studies and informal science learning, however they did not use systematic review methods, included evaluations published after 2011 only, and excluded studies of individual activities within festivals.

Attendees of health-related PE events are likely to include patients or users of health services, including families and informal carers, as well as health and social care professionals.

Given the needs of this audience and the potential demand for and interest in health-related science festivals, understanding best practice in the evaluation of these events is crucial. However, there are no published syntheses of evidence in this area.

While guidance is available for researchers evaluating a PE event [22, 23], PE evaluation efforts have been criticised for poor design, execution, and interpretation [20], for example, use of a restricted range of evaluation methods [21], and using evaluation as a token activity to justify funding [24]. The Queen Mary University of London (QMUL) public engagement evaluation toolkit [25, 26] has been developed as an open-access, pragmatic, generic toolkit applicable to diverse forms of academic PE and proposed as a "common 'evaluation standard'" [27]. The toolkit gives practical advice about evaluation methods, the adoption of which, the authors suggest, could result in more consistent and higher quality PE evaluations, offering valuable data about the impact and value of health-related engagement activities at festivals [28]. We chose the QMUL toolkit as an appropriate comparator for this review as it is familiar to health researchers [29], and is applicable to a wide variety of engagement activities, including those evaluated in the studies included in this review, which utilise multiple different PE approaches and frameworks. In this review we aimed to comprehensively synthesise the evidence from evaluations of health-related PE festivals. Our primary research question was: What methods and outcomes are reported in published evaluations of health-related public engagement festivals? Our secondary question was: How do the evaluation methods used in these reports compare to those outlined in the QMUL public engagement toolkit [25, 26]?

## Methods

We conducted a systematic review with narrative synthesis [30], to comprehensively describe and synthesise the methods and outcomes of health-related PE festival evaluations. The protocol for the review was registered prospectively on Research Registry (ID review registry 1021) [S1 File]. There were no amendments made to the protocol.

### Search strategy

The following databases were searched on 28/12/2020: MEDLINE, Embase, and CINAHL (all via OvidSP) and Web of Science—core collection, with the search restricted to publications since 1 January 2000. Literature scoping and discussion with a subject librarian helped to inform the choice of databases and the search strategy. The search strategy was adapted for each database by combining the same groups of search terms, namely, "public engagement", engagement type (i.e. "festival" or "event") and topic (i.e. "science", "research" or "health"). Search strings for each database can be found in S1 Table.

### Inclusion and exclusion criteria

Inclusion and exclusion criteria were established *a priori*. To be included in the synthesis, studies had to self-identify the evaluated event as a 'festival'; state public engagement i.e. two-way dialogue with the public [3] as one of the festival aims; provide evaluation data on adults; be a single or multi-year festival; be on a human health-related topic; and be an arts, culture or science festival which had an identified health-related theme or activity with evaluation of the health-related element. We included studies where festival audiences were members of the general public, i.e. who were non-specialists and not in academia or teaching. The following definitions of 'public engagement' and 'festival' were developed for this review to support application of the inclusion criteria:

**'Public Engagement':** Two-way dialogue between health-related researchers (including social scientists) or PE practitioners and members of the general public [3]. We focus here on

engagement in relation to health-related research or a health topic, including medicine and applied health.

'**Festival**': A live event which engages the public in health-related science or a health-related topic. The event had to be transient, provide a brief and concentrated focus on the topic, and take place in a specific place or region.

Studies were excluded if they: (1) used festivals to recruit participants for research, policy or service planning or prioritisation; (2) implemented the festival primarily as a health intervention (i.e. to bring about a change in health-related behaviour); (3) were published before 2000; (4) evaluated festivals with no health-related science or research remit/ not on a health topic; or (5) evaluated PE events which did not fit our definition of 'festival'. Searches were limited to English language reports of empirical studies published since the year 2000, since most PE festivals have emerged in the last twenty years [18].

We chose to include only reports of studies where the audience included adult participants, to ensure the festivals and their evaluations were comparable. Evaluation of the PE impact on children is often mediated by adults (e.g. teachers and parents), and uses different delivery formats, purposes, venues and times compared to adult-orientated PE events [21]. Studies of mixed populations of families/ children and adults were included if the adult data could be extracted for the synthesis. Festivals which evaluated the impact only on children or student and teacher participants were excluded.

## Study selection

Records were managed and deduplicated in EndNote [31]. Titles and abstracts of retrieved records were screened for eligibility (SM), with 2% independently assessed by a second reviewer (LS/CC). Full text screening for study inclusion was undertaken by SM, with a random 20% sample screened by LS and CC. Citation tracking and hand searching of the reference lists of included papers was undertaken to identify any further eligible papers (SM). LS and CC independently reviewed 10% of the data extraction (performed by SM) to check for refinement or omission of data, and 10% of the quality assessment. Where there was uncertainty over study eligibility, data extraction or quality rating, this was discussed between the three researchers to reach consensus.

## Data collection

A data extraction table was developed and piloted during the screening process. Data were extracted under the following headings: First author's name, report title, year of publication, location, name of festival/ event, aim of festival, aim of the evaluation, evaluation methods, evaluation outcomes, evaluation conclusions, researcher relationship to the festival, internal or independent evaluators, sample size/ response rate and total festival/ audience size. Data were also extracted specifically for appraisal against the QMUL toolkit under the headings of design, delivery and impact [25, 26] and the additional QMUL toolkit subheadings (S2 Table).

## Quality appraisal

A validated critical appraisal tool, the Mixed Methods Appraisal Tool (MMAT) Version 2018 [32] was used to assess the quality of included studies. The MMAT allows for methodological quality appraisal of qualitative, quantitative, mixed-methods, randomised controlled and non-randomised studies. As recommended in the MMAT user guide, studies were not excluded based on their quality. However, the narrative synthesis reflects and includes discussion on the quality of the included studies.

## Data analysis

A narrative synthesis of collected data was carried out following the framework stages proposed by Popay, Roberts, Sowden et al. [30]. Narrative synthesis was selected a priori because studies identified during literature scoping included a range of designs and aims, and were insufficiently similar to complete meta-analysis or meta-ethnography [33]. The framework stages used in this review were [30]:

1. Developing a preliminary synthesis

2. Exploring relationships in the data

3. Assessing the robustness of the synthesis product.

Comparison with the QMUL toolkit further refined the appraisal and synthesis of the included studies and informed recommendations.

For the preliminary synthesis, we tabulated and grouped evaluation methods and outcomes. Evaluation outcomes which were conceptually similar were grouped and data cross-tabulated based on recurring data, potential moderating factors and factors implicated by existing literature, e.g. study methodology, demographics and sample size [4, 19]. This cross-tabulation and concept mapping enabled visual representation and exploration of the data and relationships within it [33]. The strength of the evidence was examined using the quality appraisal data and consideration of bias in the included studies. Summaries and conclusions were drawn from this data interrogation.

SM led the synthesis, with regular meetings with LS and CC to review preliminary findings and patterns in the data.

## Results

Database searches identified 407 records after deduplication, with one further reference identified through hand-searching the reference lists of included studies and relevant reviews (Fig 1) [34]. Eight studies met the inclusion criteria [35–42].

Key study characteristics are described in Table 1. Six of the eight included studies were published between 2015 and 2020 [36–39, 41, 42]. Seven of the eight studies used mixed-methods research [35–40, 42], with one study using quantitative methods alone [41].

Most included studies were conducted in the UK (four in Scotland [35, 36, 39, 40], one in England [42], and one study each was from Indonesia [37], the USA [41] and New Zealand [38]. Five of the evaluations were of events embedded within larger festivals [35, 36, 39, 41, 42]. One event took place in an air raid shelter [42] and three in a performing arts space [37, 39, 40]. Two of the eight festivals were on the topic of mental health [37, 40]. One of the studies aimed to evaluate the whole festival [36], whilst the other seven studies evaluated a specific element of the festival.

### Quality appraisal

A summary of study quality is given in Table 1. Detailed study quality analysis is presented in the S4 Table. Five of the eight studies had superior methodological quality, meeting four or more of the five criteria in the relevant category for the study design [38–42]. The pure quantitative study was of high methodological quality [41]. Two studies were rated as low methodological quality due to inadequate reporting [35, 36]. Data extracted on study characteristics showed that the studies which had separate researcher or evaluation teams were of higher methodological quality [37, 38, 40–42]. One study did not explicitly state the relationship between the evaluation and festival teams [39].

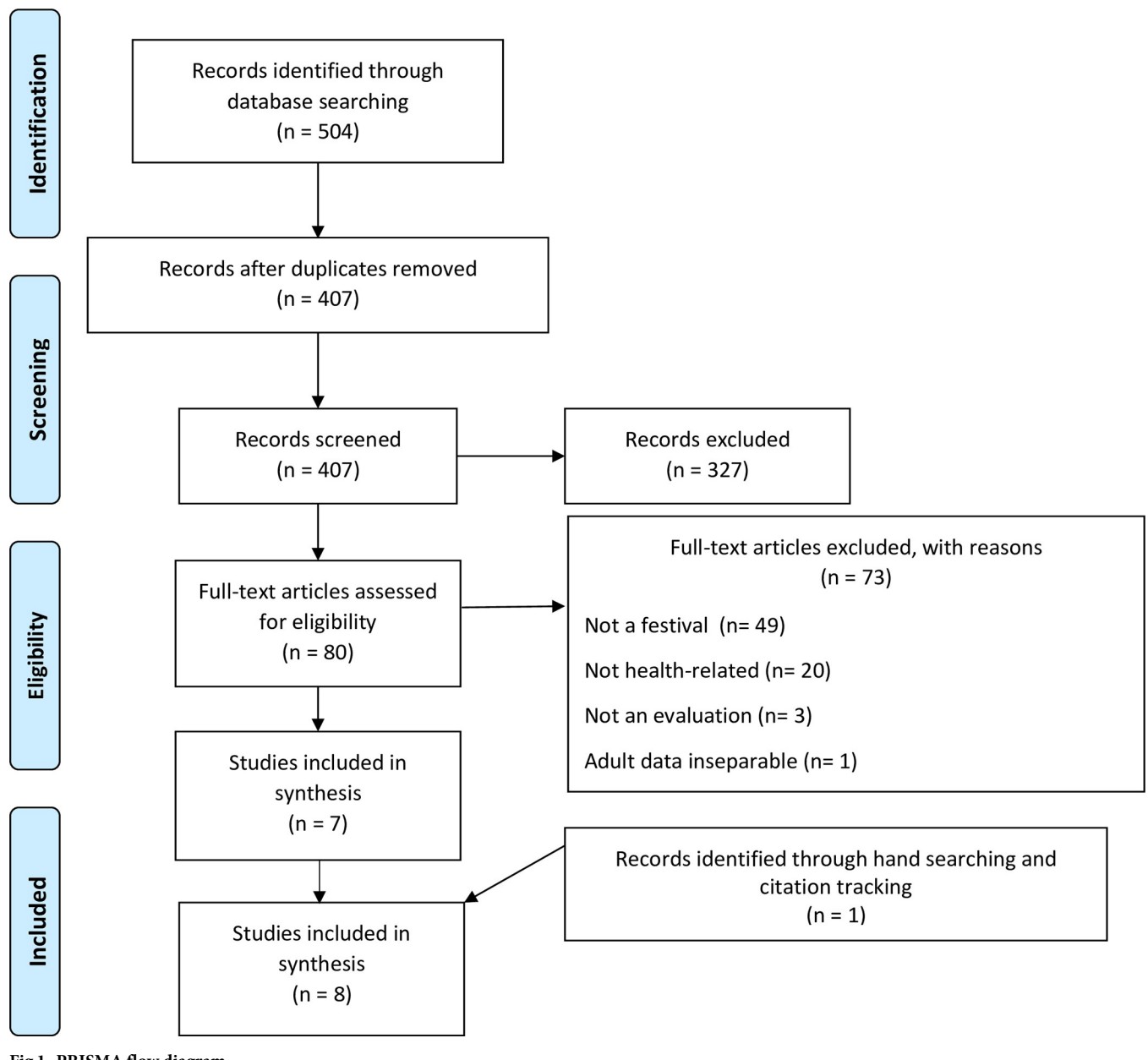

**Fig 1. PRISMA flow diagram.**

Across the mixed-methods studies, researchers frequently omitted key information to assess the quality of either their qualitative or quantitative methods [35–37, 39]. For the qualitative component, some reports did not state the theoretical position underpinning the research question and did not describe any data analysis methods [35, 36]. The quantitative aspects of the mixed-methods studies commonly underreported on their sampling strategy [35, 36]. They also sometimes failed to report missing data or its management [35–37, 39]. Examples of good quality data analysis include stating hypotheses for testing and using statistical tests to compare festival attendees to the general population [41].

**Table 1. Summary of included studies.**

| #, Author, Year | Title | Location | Audience size | Aim of festival / Event | Evaluation aim | Evaluation methods and Researcher relationship to festival / events | Evaluation sample size (response rate) | Outcomes assessed | Evaluation conclusion | Summary of MMAT quality appraisal |
|---|---|---|---|---|---|---|---|---|---|---|
| (#1) Brooks, 2019 | "Evaluating the acceptability of a co-produced and co-delivered mental health public engagement festival: Mental Health Matters, Jakarta, Indonesia" | Jakarta, Indonesia | 737 attendees over the 6-day festival | "1. To improve knowledge of mental health amongst attendees through a co-designed and co-delivered mental health festival 2. To strengthen relationships between community organisations, health services and higher education institutes and explore the potential for future festivals 3. To promote future engagement in mental health research" | "To explore the impact of the festival on knowledge/ understanding of mental health and future behavioural intentions 2. To develop understanding on the acceptability of undertaking mental health festivals in Indonesia to raise awareness of mental health." | Mixed-methods: post-event structured and unstructured evaluation forms. Researchers were separate to the festival organising committee. | Forms distributed to all attendees 324 attendees completed forms (43.9% response rate) | QUANT: Design (marketing/ advertising (how did you hear about the event) Acceptability: why did they attend, Delivery: Demographics (age gender, role) experience; Which events did you attend? Quality of festival and festival overall. Values: relevance to life/ times. Emotional engagement (did you feel moved or inspired/ engaged in experience). Understanding and knowledge (increased understanding of topic). Attitude: want to find out more, exposed to new ways of thinking, intention to be involved in research. QUAL: Values: important/ relevance. Understanding: enhanced understanding via speakers, venue, community, and service users (credibility). Emotional engagement: with community and experts by experience. Engagement: via arts-based/films. Experience: improvement, size of venue and event, publicity, event duration and frequency. | A co-produced art-based mental health festival is an acceptable way to increase understanding of emotional health issues / promotion of public mental health to Indonesian populations. | Good quality: Qualitative data higher quality than quantitative data. Acknowledged self-selecting sample. No reporting on rationale for using mixed-methods design and limited reporting of study limitations e.g. no discussion of impact of missing data on results, and no comparison of festival population to sample. |
| (#2) McCauley 2019 | "BIRTH: a mixed-methods survey of audience members' reflections of a global women's health arts and science programme in England, Ireland, Scotland, and Switzerland" | Edinburgh, Scotland | Estimated total attendance based on theatre capacity. | "To raise awareness and debate regarding global inequalities in access to, availability and quality of maternal healthcare." | "To assess the views and experiences of audience members who had just watched a play and/ or been involved in an expert panel discussion as part of the BIRTH programme." | Mixed-methods: post-event questionnaire with structured and open questions. It is unclear if the researchers were part of the BIRTH multi-disciplinary team or separate. | 176 respondents in Edinburgh, 17 responses for the extended questionnaire 159 to the short questionnaire. (42% response rate, estimated All audience members asked to complete a questionnaire. | QUANT: Demographics (age, gender, ethnicity, theatre views) Experience: Emotional engagement: emotionally moving, identified with the characters, felt challenged and provoked. Attitude: me think differently, concern about the topic, wanting to find out more. Knowledge and understanding/ learning: learnt something knew, opened eyes to new ideas. QUAL: Free text 5 themes: Thanks/ positive feedback (views), Innovative use of arts and science (experience), Personal feelings (engagement), Need for action (views), Suggestions for use in schools/ education (views). | "The BIRTH programme can be used as an effective tool to engage the public/stimulate debate, deliver key messages, and raise awareness of [these/ global women's health] issues". | High quality. Appropriate use of thematic analysis and coding of qualitative data, with data quoted in results under themed headings. Clear discussion of limitations including of estimated response rate, self-selection bias and convenience sampling strategy, but missing data not discussed. |
| (#3) Verran, 2018 | "Fitting the message to the location: engaging adults with antimicrobial resistance in a World War 2 air raid shelter" | Stockport, England | 37 (out of 40 spaces available —sold out on Eventbrite) | Event about AMR designed activities to address 5 questions which framed the event. "i) How important are antibiotics to us today? ii) How did we cope without them? iii) Can we find new alternatives to antibiotics? iv) Can we develop alternatives to antibiotics? v) Why is AMR an issue and what is being done to address it?)" | "To develop, deliver and evaluate an event designed to engage an adult audience with anti-microbial resistance." | Mixed-methods: activity output and observation Researchers involved in event planning and the "lead author introduced the audience to the event." | 37 (35 gingerbread men returned, 62 agar plates / swabbed and Flickr access 57 times.) | Engagement with activities QUANT: frequency counts of activity output (gingerbread men, agar plates, Flickr access) (not pre-specified: percentage of bacteria names recognised) and QUAL: recording questions and observation of level of engagement | "Hands on practical engagement with AMR can enable high-level interaction and learning in an informal and enjoyable environment." | High quality. High response rate, clear description of appropriate data collection methods. Qualitative and quantitative data were integrated and rationale for mixed methods given. Some findings not supported by data. Missing data is minimal. |
| (#4) Rose, 2017 | "Engaging the Public at a science festival: Findings from a panel on human gene editing" | Wisconsin, USA | 125 people attended the panel. | The effects of an engagement activity on audience perception of controversial science topic. "1. To increase participants' basic knowledge about human gene editing. 2: increase both risk and benefit perception related to human gene editing, therefore increasing participants' understanding of the complexity of the issue and potentially avoiding polarization based solely on moral concerns." | "Explore 1. if the Wisconsin Science Festival human gene editing panel increased familiarity or perceived knowledge levels. 2: if participating increases risk perception 3. if participating increases benefit perception. 4. how will the panel affect attendees' moral and ethical views of the technology." | Quantitative pre-post structured survey. Researchers separate to the organisers and delivery personnel. The festival "is organized by public and private institutions". The panel "were all university faculty members". | 34 responses to the pre-test survey (94.1% response rate). 26 responses to the post-test survey (100% response rate). "Randomised selection… every 5th person to enter/ leave plus $2 incentive. (16 people engaged with the panel during the discussion period)." | QUANT: reach (demographics to see if the samples before and after were the same or not: age gender degree) Knowledge (perceived knowledge level before/ after), Attitude (risk and benefit perception and x1 ethics question and x2 morals questions all before/ after) | Attendees felt an increase in "perceived knowledge levels, risks perceptions, benefit perceptions and moral acceptability". | High quality: Detailed discussion of limitations e.g. acknowledgment of small sample size, and strengths e.g. using randomised sampling, statistical comparison between pre- and post- samples and between audience and wider population. |

(*Continued*)

**Table 1.** (Continued)

| #, Author, Year | Title | Location | Audience size | Aim of festival / Event | Evaluation aim | Evaluation methods and Researcher relationship to festival/ events | Evaluation sample size (response rate) | Outcomes assessed | Evaluation conclusion | Summary of MMAT quality appraisal |
|---|---|---|---|---|---|---|---|---|---|---|
| (45) Brookfield 2016 | "Informal Science Learning for Older Adults" | Edin-burgh, Scotland | 50 people registered for stand-alone event and 40 people registered for the festival event. | "A free-to-attend science festival style event that presented "content" linked a project (MMP) and that incorporated several different learning formats and engagement techniques in order to gain insights into what worked." (As a stand-alone event and within the Edinburgh International Science Festival). | "Report on the process involved in creating and promoting the event and the overall experience of delivering it in two different settings." "To reflect on where the event succeeded and how it could have been improved, and consider its performance as a vehicle for older adults' learning." | Mixed-methods: post-event audience feedback forms and commentary Researchers developed, delivered and evaluated the event. | 38 forms in total. 39 people attended the stand-alone event. 18 people attended the festival event. | QUANT: number registered/ attended. Demographics: Age. Experience: Enjoyableness / usefulness of events Knowledge: would attendees share / use what they learnt. Attitude: Would they attend again. Behaviour: distance travelled. Marketing: Tweets. QUAL: Behaviour: previous event attendance, levels of participation, mix of activities, favoured activity, questions asked. Experience: Engagement (with event and researchers), satisfaction with venue, Marketing: researcher labour time and costs. | "There is appetite for informal science learning among older adults" This type of event might be an "appealing and appropriate vehicle for informal science learning" but access can be restricted by venue choice by using an established science festival to support administration and marketing | Low quality: Unclear and unreported approach to qualitative data collection and analysis. Not clear which data supports the conclusions or conclusions unsupported by data or quotes. Unclear sampling strategy and missing data not reported. No rationale for using a mixed-methods design. |
| (46) Fogg-Rogers, 2015 | "Knowledge is Power": a mixed-methods study exploring adult audience preferences for engagement and learning formats over 3 years of a health science festival." | Auckland, New Zealand | Approx. 3000 attendees at the event each year, | "To communicate information about brain health and disease along with current neuroscience research, while also engaging publics in the ongoing research process." | Explore audience preference for engagement styles at science festivals Research question: "what formats do audiences at a science festival prefer and why?" | Mixed-methods: post-event questionnaire with structured and open questions Researchers separate to festival organisers/ delivery team. Festival is a part of "nationwide events coordinated by the Neurological Foundation of New Zealand." "The event is staffed by volunteer neuroscience researchers and students". | 661 returned over 3 years (annual response rate approx. 7%) with mean 220.3 returned each year (SD = 24.6). Aiming for a cross-sectional sample. | QUANT: Demographics: age gender ethnicity and education levels. Living with a brain disease (not pre-specified). Experience: audience format preference via a x3 Quant Q's; perceived attractiveness, perceived usefulness, and Behaviour: attendance. Likert Scales: Experience was it a good day out for the family; Knowledge and Understanding: helped me learn more, lectures are a good way to get info, I did not learn anything, I cannot understand neuroscience). QUAL: Open questions on why they had certain preferences / rankings of formats: data grouped thematically: Interested in learning (knowledge), Knowledge is power (knowledge), Career and professional development (knowledge), Research and expert opinion (experience), engaging in curiosity (experience). | "Health Literacy as an Asset: Festival formats employing traditional public understanding of science style communication, namely lectures, were preferred by the majority of adult participants, with the primary learning. Lectures in an asset-based model means expert dissemination of research findings are central to an engagement model, building on the knowledge, skills and understanding that people already hold." | High quality: clear rationale for using mixed methods, to triangulate data, questionnaire was piloted, description of handling of missing data given, thematic analysis of qualitative data explained and quotes used in results under themed headings. Discussion of limitations including acknowledgment of potential sampling and response bias. |
| (47) Bird, 2013 | "Getting Science to the Citizen—"Food Addiction" at the British Science Festival as a Case Study of Interactive Public Engagement with High profile Scientific Controversy," | Aberdeen Scotland | Over 170 attended the workshop/ sold out in advance | "The event addressed the controversial and high-profile area of 'food addiction'," "to engage the public in dialogue about science." | Not stated only: "the event was evaluated"/ to describe the event | Mixed Methods: post-event feedback form. Not explicitly identified as evaluation methods: voting buttons, show of hands (frequency counts/ %) and an interactive challenge. Researchers initiated and presented the event. | 121 completed forms. | Feedback Form: QUANT: Demographics, Experience: enjoyment and interest, met expectations, Knowledge: learnt something new, Values: relevant to life. Show of hands/ interactive challenge (no data reported), voting buttons (knowledge, activity output/ engagement). QUAL: what did you enjoy? Experience: interactive, presentation, accessible/ pacing. Knowledge: information, interesting. Subjective researcher reflection: Engagement e.g. researcher perceived shock and resonance in the audience. | "There is public appetite for events related to real-life health issues, the event was a successful formula for controversial topic'." "Audience appeared receptive to new information and clarification.""It was a positive experience for presenter and audience." | Low quality: Vague research question. Unstated methods for qualitative data analysis. Some results unsupported by data. Sampling strategy unreported. Missing data unreported. No stated reason for using a mixed-methods design. Data insufficiently explained to comment on divergences in the data. |
| (48) Quinn, 2011 | "The impact of a national mental health arts and film festival on stigmas and recovery" | Glasgow and Lan-arkshire, Scotland | 3000 at the festival in total, 1318 at the evaluated events. Attendance at events ranged from 3 n = 15–113 | To challenge stigma and discrimination against people with mental health problems: 1) to promote positive attitudes towards mental health amongst opinion formers and the public through arts and culture; 2) To strengthen the links between arts, community and public organisations, and explore the evidence and support for an annual festival. | i) To identify who might attend mental health arts events; ii) To identify the impact upon knowledge, attitudes and likely future behaviour; iii) To explore whether specific components of stigma (e.g. social distance, perceived dangerousness, possibility of recovery and unpredictability) were influenced by specific arts events; iv) To learn lessons for developing an evaluation framework for complex events in real-life circumstances. | Mixed Methods: Quantitative pre- and post-event questions and qualitative post-event questions on evaluation cards Festival organised by multiple organisations, and one of the researchers worked with one of these organisations (Mental Health Foundation) | 20 out of 31 events evaluated, 415 respondents out of 1318 who attended the 20 events. Response rates ranged from 9.7%–63.6%. | QUAL Experience: Felt Inspired, reflection on attendee's own mental health, role of the arts is important, enjoyment. Attitude: Acceptance (of difference/ society), need for support, behavioural intent (won't change behaviour), will act positively, will participate in arts, will change work practices, will change personal health-related behaviour, will be an activist). Knowledge: Of mental health (understanding of recovery, understanding of mental health perspectives, awareness of social factors/ opened eyes, understanding of different mental health perspectives, understanding impact of mental health). Behaviour: numbers of attendees. QUANT: Demographics (age, gender), Attitude: (baseline attitude (pre-event), change in attitude pre-post event), | Modest evidence that an arts festival can impact stigma. "A collaborative national arts festival can contribute towards reducing stigma and should integrate with other national initiatives that address stigma and promote public mental health." | High quality: Reason for use of mixed methods clearly stated and explained, detailed discussion of limitations including justification of use of opportunistic sampling method, selection effects and ceiling effects. Missing data reported in results. |

**Table 2. Evaluation tools utilised.**

| Evaluation Tool | Record ref number | | Total number of studies |
|---|---|---|---|
| | **Higher quality** | **Lower quality** | |
| Structured post-event self-completion questionnaire | #2, #6, #8 | #1, #5, #7 | 6 |
| Open question post-event self-completion questionnaire | #2, #6, #8 | #1, #5, #7 | 6 |
| Structured pre-event self-completion questionnaire | #8 | - | 1 |
| Festival activity output | #3 | #7 | 2 |
| Spoken audience questions (recorded) | #3 | - | 1 |
| Observation of engagement | #3 | - | 1 |
| Structured pre-event survey (administered) | #4 | - | 1 |
| Structured post-event survey (administered) | #4 | - | 1 |
| Social media analytics | - | #5 | 1 |

## Evaluation methods

The most prevalent method of evaluation (n = 6/8) was a self-completed post-event question-naire, with structured and open questions [35–40] (Table 2). Studies with large sample sizes used questionnaires, while the two studies with the smallest evaluation samples used more labour-intensive evaluation methods, e.g. observation [42] and in-person surveys [41] (see supplementary material). These latter studies also had separate evaluators collecting the data, higher response rates and were of high methodological quality. Two studies of higher quality also collected pre-event data [40, 41], while another used an electronic voting system for the audience [39]. Two studies without separate evaluation teams had broad or unspecified evalua-tion aims and poorer methodological quality [35, 36]. Studies of higher methodological quality used a wider range of evaluation methods [38, 40–42].

## Evaluation outcomes

Evaluation outputs and outcomes (as defined by Grant (2011) [43]) were grouped into four conceptual themes: reach, attitude, knowledge and experience (Fig 2). Four of the studies eval-uated outputs/outcomes in all four themes [37–40]. One study exclusively evaluated the attendees' experience [42]. Reach, knowledge and experience were assessed by seven out of eight studies [35, 36, 38–42] and attitude by six out of eight [36–41].

The studies often used the terms 'participant', 'audience', 'visitor' and 'attendee' somewhat interchangeably to describe the people involved in the festival activity. Although the term 'audience' might indicate a more passive level of engagement (e.g. just listening) and 'partici-pant' a more active style of engagement (e.g. sharing opinions), these studies generally did not define such terms.

## Reach

All except one evaluation [42] assessed participant age; five out of eight assessed gender [37–41] (Fig 2). More women than men attended the festivals [41] and completed the evaluations [37–40] (Table 3). Only two studies reported demographic data on ethnicity [38, 39], with the largest proportion of participants self-identifying as "white" [39] or "New Zealand/ European descent" [38]. Data on attendee education level [38, 41] or occupation [37], though only mea-sured in three studies, indicated that visitors represented in the evaluation samples were largely well educated.

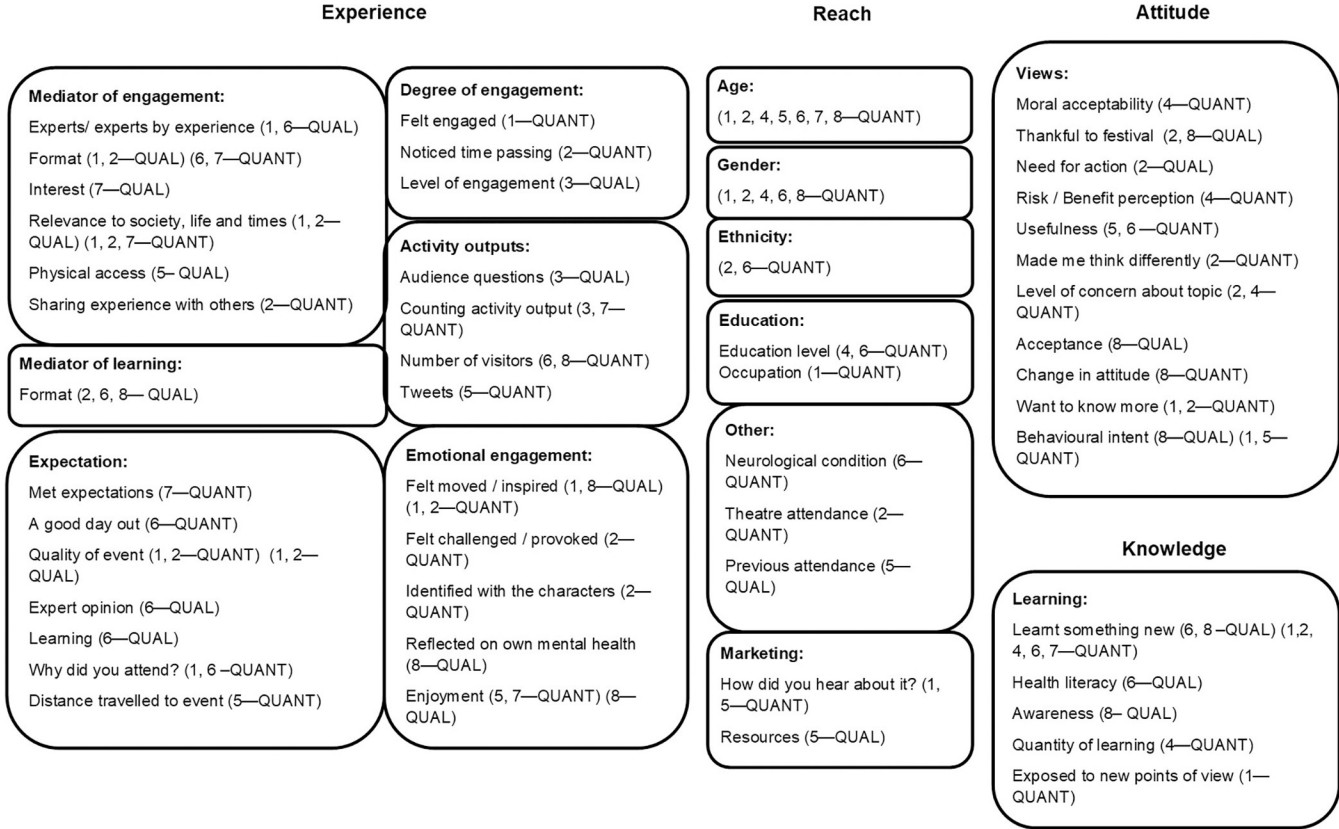

**Fig 2. Conceptual map of evaluated outcomes and outputs.**

Only two studies reported on the marketing of their public engagement event [36, 37] with one study including social media analytics as part of their marketing assessment.

## Attitude

Of the six studies which evaluated the attitude of attendees, only two measured this using a pre-post quantitative methodology [40, 41]. Both these studies had high methodological quality. All other attitude outcomes were evaluated post-attendance. Quantitative methods were frequently used to evaluate attitude, with only two studies adopting qualitative methods, although both these studies were of high methodological quality [39, 40]. Three studies looked at attitudinal outcomes involving attendee behavioural intent [36, 37, 40]. Two studies evaluated four or more outcomes related to attitude [39, 40].

## Knowledge

Outcomes related to audience knowledge were evaluated quantitatively in five studies [35, 37–39, 41] and qualitatively in two [38, 40]. Only one study, of high methodological quality, used mixed methods [38]. Evaluations commonly asked attendees after the event to indicate if they felt they had learnt something new. Five of the six studies evaluating knowledge were of high methodological quality [37–41].

**Table 3. Demographic data for studies reporting reach.**

| Record ref number | Demographics of evaluation sample or entire festival population. | Gender | Age (years) | Ethnicity | Occupation | Education |
|---|---|---|---|---|---|---|
| **#1** | Evaluation | female = 88.6% (n = 286) Male = 10.8% (n = 35) | Mean = 22.5 years Range = 17–51 years | n/a | Student = 50.3% (n = 163) Patient/ public = 20.4% (n = 66) Professional = 5.3% (n = 17) Missing 24.1% (n = 78) | n/a |
| **#2** | Evaluation | Female = 12 (71%) Male = 5 (29%) | 18–30 = 1 (6%) 31–40 = 1 (6%) 41–50 = 3 (18%) 51–60 = 7 (41%) >60 = 5 (29%) | Asian/ Asian British = 0 Black/ Black British = 0 Mixed = 0 Other = 0 White = 17 (100%) | n/a | n/a |
| **#4** | Festival | Female = 52% | Median = 41 years | n/a | n/a | College degree = 91% |
| **#5** | Evaluation | n/a | >65 (n = 21) (out of 38 people who completed the evaluation form) | n/a | n/a | n/a |
| **#6** | Evaluation (data aggregated across three years) | Female = 66.4% | Range = 7–87 years Means age = 48.5 years 50–64 years = 25.5% (dominant age category) | New Zealand European descent = 64.1% Asian = 11.2% Maori = 1.9% Pacific Islanders = 1.7% | 16% of respondents citing the festival was relevant to "career path or job" e.g. updating professional knowledge, directing future career path or the knowledge gained would be useful in their profession. | Post-graduate studies = 42.3% Undergraduate education or a trade certificate = 25.2% No formal education post-secondary school = 26.6% |
| **#7** | Evaluation | n/a | "majority 19–40 years" | n/a | n/a | n/a |
| **#8** | Evaluation | In evaluation sample: Gender ratios per event reported (two events with all-female audience) In festival: Male = 30.8% / Female = 69.2% | Average age and range of ages per event reported. Average ages per event ranged between 25.2 and 71.9 years, audiences ages ranged from 13–89. In Festival: "higher proportion of younger people, (especially those between 25 and 34 years) than the Scottish population" given as percentage per age range. | n/a | n/a | n/a |

Note: study #3 [42] did not report demographic characteristics

## Experience

All studies except one [41] assessed audience experience, with multiple indicators used (Fig 2). Audience experience was commonly measured through engagement-related outcome or output e.g. emotional engagement, degree of engagement, mediator of engagement such as format. Engagement was evaluated by all except one of the seven studies which evaluated experience [42]. Attendee emotional engagement was assessed by five of these seven studies [35–37, 39, 40].

Characteristics which were reported to enhance engagement at health festivals included the format (e.g. theatre, film, lecture) [35, 37–39] and the use of community, patient, or research experts [37, 38]. Three studies reported on perceived relevance of the content to attendees' life, society or times and whether it met participant expectations [35, 37, 39]. Five studies recorded

and evaluated audience outputs, such as audience questions and counting the number of visitors [35, 36, 38, 40, 42]. One study used the physical output from activities within the festival as a more objective measure of engagement, e.g. number of materials used or submitted [42]. This study also measured the degree of engagement qualitatively, using observation.

## QMUL toolkit

The conceptual map in Fig 2 indicates whether the studies were evaluating design, delivery/outcomes or longer-term impact (e.g. evaluation activities sometime after the original project is completed), as specified in the QMUL toolkit [25, 27]. 7/8 studies (with [42] the exception) evaluated festival design; all studies evaluated festival delivery, and no studies evaluated long-term impact. One study used aggregate data across three years [38]. The QMUL toolkit offers a range of 21 different tools to use for evaluation [26]. Evaluation methods described in the toolkit and applied in the studies included 'structured questionnaires', 'public lecture multiple-choice questions' and 'event feedback forms' [35–41]. Additional methods employed in the included studies, but not listed in the toolkit, included frequency counts of physical outputs from the festival and observation of audience behaviour [35, 42].

## Discussion

In this systematic review of PE health-related festivals, eight studies were eligible for inclusion, and all were published in the last ten years, despite the science festival scene burgeoning over the last two decades [18]. PE was evaluated predominantly via mixed-methods, often using self-report questionnaires. Evaluated outcomes included reach, experience, knowledge and attitude. A limited range of evaluation methods were used compared to an existing evaluation toolkit, and no long-term outcomes were evaluated.

The studies' frequent use of evaluation forms is in line with observations of overreliance on audience self-report [21]. However, the higher quality studies used a wider range of evaluation methods. Researchers are encouraged to use technology-based and unobtrusive evaluation methods [21], to reduce feedback burden on attendees and provide alternative ways to capture data [18, 44]. One of the studies we included used social media data [36] and another avoided using an evaluation form completely, in favour of observation and frequency counts of activity output [42]. Evaluators must ensure though that their methods are appropriate as certain evaluation aims require specific research methods; for example, measuring changes in audience attitudes requires a pre-post design as a minimum standard [20], but of the six studies which assessed attitude, only two studies used this design [40, 41].

The evaluations collected outcome data on attendee reach. Existing literature highlights that science festival audiences are often biased towards people already interested in science and rarely have good representation of minority ethnic groups [19, 24, 45]. For health-related PE festival evaluations, the 'already interested' population includes health professionals. Indeed, one study indicated that visitors attended the festival to further professional knowledge or career development [38]. Festivals often target schools and families as part of formal education or to capitalise on those already interested in science [19, 46]. However, families from more deprived areas and parents or adults without degrees are less well represented at festivals [21, 44–46]. Given these known biases, it is concerning that reach was not more thoroughly evaluated in the studies we identified. Audience members in our studies were mostly female, well-educated, and ethnically white, indicating a gender bias and lack of ethnic diversity in the evaluation samples. Coupled with the high education status of the attendees, this indicates the festivals included were restricted in their reach. It is encouraging that one study

discussed improving reach as a future aim [39] and another specifically attempted to reach an under-represented older age group [36].

The literature suggests that science festival audiences value interaction with scientists or experts [21, 24]. The evaluation data we identified corroborate this finding and suggest further that learning and engagement are positively mediated by contact with experts by experience, as a result of patients and other health service users being involved in the delivery and design of the PE events [37, 38]. For example, patient stories may play an important role in audience engagement [40]. Since involving community partners or patients is a distinctive feature of human health-related festivals, more research is warranted to establish how and why the public can best influence festival design, delivery and impact. Alongside existing literature [47], data on engagement from co-produced events [37] and events tailored to specific audiences [36], could provide useful guidance to other science festival organisers on how to include and engage a diverse audience.

It was important that the studies evaluated audience attitude because PE with research can involve ethically challenging discussion [41] or be concerning to the public [39]. One study acknowledged the responsibility of festival organisers to provide adequate reassurance or support to audience members who are engaging with emotive topics [39] and another study discussed how a dialogue-based delivery format enabled conversations about attitude [41]. A responsibility for audience well-being and the impact on audience attitude is particularly important for health-related topics, where attendees may be personally affected. This is exemplified by both studies on mental-health topics which evaluated audience attitude [37, 40].

It could seem regressive that all but two of our studies assessed an outcome related to the knowledge or learning of the attendees [36, 42], because the literature notes that science festivals have moved from informing audiences to actively engaging them [19]. However, in addition to knowledge, the studies also evaluated a range of other outcomes, e.g. experience and attitude. This suggests that, as recommended [21, 24], health-related PE festivals are not just unloading knowledge onto a passive audience via what is known as the 'deficit model' [48], but are using the two-way element which festival interaction enables to achieve multi-dimensional impact. Fogg-Rogers et al. (2015) [38] argue that, uniquely for health, knowledge gained at a festival could improve health literacy. This is in line with Ko's (2016) [48] view that knowledge outcomes are still required to ensure factual comprehension is accurate, thus helping to prevent any negative physical or legal health-related consequences. Whilst health-related festivals are trying to be more dynamic in their evaluations, there is still a place for evaluating knowledge.

All except two of the festival evaluations were published before the QMUL toolkit [27], and none of the studies were informed by the toolkit. Evaluations universally assessed the design or delivery of their festivals, with none assessing longer-term impact. Established alternative and creative qualitative data collection methods are listed in the QMUL toolkit, such as interviews and focus groups, and the toolkit also offers some technology-based evaluation method ideas, e.g. mobile event app, aerial photography [26], but these were not evident in the range of methods used by the studies in our review.

It is important for evaluators to clearly define and differentiate immediate evaluation outcomes and outputs from any other impacts for clarity. This clarity can be supported through consistent use of terminology [27]. At present, terms are used inconsistently: for instance, Quinn et al. 2011 [40] evaluate the "impact" on stigma by assessing attendee attitude immediately after the event, while Verran et al. 2018 [42] use audience engagement, assessed via outputs and observations, as an indicator of impact. Whilst strict adherence to the QMUL toolkit could restrict the creative development of evaluations [21], application of the toolkit, including adoption of its terminology, might improve individual evaluation quality, increase learning

derived from each festival and facilitate comparison. We therefore recommend that PE evaluations clearly define the concepts being evaluated; the QMUL toolkit may provide a useful reference point in this regard. Related to this, evaluations would benefit from more explicit discussion of the aims, framework and assumptions underlying PE initiatives. Differences in conceptualisation and fundamental approach (e.g. regarding the role of the public in the engagement experience) have implications for the choice of appropriate outcomes and evaluation methodology [8, 49, 50].

Assessing the longer term and broader impacts of festival activities can be practically difficult within a time-limited research grant, but more reflective opinions of a festival and accounts of whether, for example, potential changes in behaviour translated into actual changes in behaviour, might still be relevant, especially when PE festivals are ongoing [18, 21, 27, 51]. Such longer-term evaluations could help explain the complex effects and interactions at play and help develop a better understanding of active ingredients and mechanisms of action in PE via festivals. One of the studies evaluating behavioural intent acknowledged that future research could address whether attendees followed through with their intentions [37]. Three other studies also discussed the need for longitudinal follow-up to their evaluations [39, 40, 42]. Health-related PE festivals are still relatively rare, which might account for the paucity of longer-term evaluations. However, alongside our finding that not all studies had separate evaluation teams, underinvestment and limited evaluation resources might also account for the lack of impact evaluations in the literature.

We found that the higher quality studies which used specific evaluation aims, a wider range of methods [41, 42] and pooled data [38] all had separate evaluation teams. This supports findings that the paucity of ringfenced time and resources for PE evaluation has a detrimental effect on evaluation quality [18]. Better resourced evaluation teams enable alternative and more rigorous evaluation methods to be planned and deployed, and independent evaluators might have more evaluation expertise than is present in a PE event team.

A strength of this review is the use of established narrative synthesis methods, which enabled the mixed-methods findings to be combined conceptually, overcoming methodological differences [30, 33]. Limitations related to the resources available for the review include not searching the grey literature, using only adult data, restricting the review to English language study reports only, and requiring that authors self-identified their PE as a festival. There might be relevant records which have not been identified in this study, particularly as, in this nascent field, terminology is not always used consistently and not all evaluations are published in peer-review journals. Using additional methods to identify unpublished studies for inclusion may have resulted in further studies for inclusion, including studies on a wider range of health-related topics. It can also be difficult to discretely categorise a PE activity from an intervention, particularly in health-related topics, however by clearly defining and reporting our inclusion and exclusion criteria, we have demonstrated transparency in our methods.

The results of this review enable us to make some recommendations to evaluators of future health-related PE and suggestions for future research. Given the need to broaden the reach of PE events and improve inclusion, particularly from underrepresented or minority groups [19, 21], evaluations should include a range of demographic indicators, including ethnicity, gender, occupation and a measure of socio-economic status/deprivation level. We found that patient/ service user involvement in event delivery supported learning and engagement [37, 38]. With the increasing focus on co-producing and co-delivering health-related PE events with patients and communities, there is a need for future research to understand and assess how the public can best influence festival design, delivery and impact. Health-related PE festivals should deliver evaluations which use consistent terminology and high-quality methodologies. Evaluators should be creative in their use of evaluation methods and open to considering a variety of

different outcomes, depending on the aims of the festival and the evaluation. Using the QMUL evaluation toolkit [25, 26] might help researchers to achieve this. Consideration should also be given to the use of independent evaluators with specific expertise and distance from the PE event. The current lack of assessment of long-term impact highlights the need for more investment into PE evaluation, which should include comparison of the impact of different PE methods as well as optimisation of PE evaluation methods.

In conclusion, whilst there are examples of high-quality reports and creative data collection methods, there is still a need to address the reach of health-related PE events and improve PE evaluation. The QMUL evaluation toolkit [25, 26] may help improve the consistency and quality of evaluation methodology and reporting. More robust evaluation of PE festivals could help to improve our understanding of how to engage with every part of a community and give clarity about which design and delivery methods work for which topics and audiences, and how best to improve reach and impact.

## Supporting information

**S1 Checklist. PRISMA-P 2015 checklist.**
(DOCX)

**S1 File. Registered protocol.**
(PDF)

**S1 Table. Search strategy.**
(DOCX)

**S2 Table. Queen Mary University of London (QMUL) toolkit headings.**
(DOCX)

**S3 Table. Study sample size, response rate and evaluation method.**
(DOCX)

**S4 Table. Quality assessment using the Mixed-Methods Appraisal Tool (MMAT).**
(DOCX)

## Acknowledgments

We would like to thank Sarah Herring at the University of Bristol for her assistance with the search strategy.

## Author Contributions

**Conceptualization:** Charlotte Chamberlain, Lucy E. Selman.

**Formal analysis:** Susannah Martin, Alison Rivett.

**Investigation:** Susannah Martin, Charlotte Chamberlain, Alison Rivett, Lucy E. Selman.

**Methodology:** Charlotte Chamberlain, Alison Rivett, Lucy E. Selman.

**Project administration:** Susannah Martin.

**Supervision:** Charlotte Chamberlain, Lucy E. Selman.

**Visualization:** Susannah Martin.

**Writing – original draft:** Susannah Martin.

**Writing – review & editing:** Charlotte Chamberlain, Alison Rivett, Lucy E. Selman.

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
