## [Decision Letter · Decision Letter 0]

1 Dec 2021

PONE-D-21-31416How are public engagement health festivals evaluated? A systematic review with narrative synthesisPLOS ONE

Dear Dr. Selman,

Thank you for submitting your manuscript to PLOS ONE. After careful consideration, we feel that it has merit but does not fully meet PLOS ONE’s publication criteria as it currently stands. Therefore, we invite you to submit a revised version of the manuscript that addresses the points raised during the review process.

We look forward to receiving your revised manuscript.

Kind regards,

Professor Benjamin Tan, BNSc MMed PhD RN

Academic Editor

PLOS ONE

Journal Requirements:

Additional Editor Comments:

Thank you for submitting your manuscript “How are public engagement health festivals evaluated? A systematic review with narrative synthesis" to PloS ONE. The editorial team have assessed your submission and a few concerns have been raised regarding the precision about terms used in the paper. Please see the reviewers’ comments for further details about necessary revisions.

Reviewers' comments:

Reviewer's Responses to Questions

**Comments to the Author**

1. Is the manuscript technically sound, and do the data support the conclusions?

Reviewer #1: Yes

Reviewer #2: Partly

2. Has the statistical analysis been performed appropriately and rigorously? 

Reviewer #1: Yes

Reviewer #2: Yes

3. Have the authors made all data underlying the findings in their manuscript fully available?

Reviewer #1: Yes

Reviewer #2: Yes

4. Is the manuscript presented in an intelligible fashion and written in standard English?

Reviewer #1: Yes

Reviewer #2: Yes

5. Review Comments to the Author

Reviewer #1: Paper Two- How are public engagement health festivals evaluated? A systematic review with narrative synthesis?

Authors: Susannah M., Chamberlain C., Alison R., Selman L.E.,

Overview:

This is relevant topic of interest to public health practitioners and researchers in the field of public engagement. Nonetheless, I suggest some substantial reforms to the authors. They are at liberty to address these suggestions to improve the readability and quality of the manuscript.

Topic:

The authors may consider changing the term “systematic review” to “Scoping review”. Mostly systematic reviews are followed with meta-analysis, whiles usually scoping reviews adopt narrative synthesis.

Abstract:

Line 13, author may specifically state the required “findings and recommendation” that may help improve future evaluations

Search strategy:

The authors may highlight the justification of searching the listed databases

Conclusion:

The authors may highlight certain areas that may require future research concentration based on the review conducted on Public Engagement festivals

Reviewer #2: Thank you for writing this paper and inviting me to comment on it. I found it a clear paper to read and thought the research was interesting, informative and at times surprising.

My criticisms of the paper are not in the research itself, but are in the framing of the paper and lack of real precision about terms used in the paper, which then has implications for the interpretation of the data and conclusions.

The argument for looking at health-related science festivals starts from a public engagement with research (PER) frame, with festivals being a potential site of PER, and health-related science festivals being a subsection of festivals and being worthy of investigation. I am not convinced that this is the most robust case for looking at health-related science festivals. I was surprised, for example, that there was no mention of the long-standing practice of PPI in health research and how PER at festivals relates to PPI. An alternative argument would be to start with science festivals in general, how they can be a site for PER, and for health PER specifically. There are others!

In the paper there are mentions of “audience experience”, “informal science learning”, “visitor studies”, “health literacy”, alongside “public engagement with research” but there is little demonstration of an understanding of the differences and similarities in evaluation that arise from these different frames. An understanding of these different areas of practice would have helped with interpreting the data. In particular there were three areas where this came through: how publics are presented, why the QMUL toolkit was used, and the observation of there being no longitudinal work presented:

1. In the paper we encounter terms such as “general public”, “lay citizen”, “patients”, “users of health services”, “professionals”, “citizens” and “audiences”. There is a substantial body of work that examines what we mean when we use these terms, for example we would largely argue that there is no such thing as a “general public”. We can describe how publics form or respond to particular interventions or circumstances. We can think about what the publics’ roles are in the engagement experience: are they audiences there to listen? participants with experience / insight to share? We can think about current levels of knowledge / attitude / behaviour with respect to the topic under discussion. We can refute the idea that all publics are citizens. This lack of critical engagement with these concepts, particularly in the different areas of evaluation practice mentioned in the previous paragraph, comes through repeatedly in the introduction and therefore in the interpretation of the data (particularly with respect to what outcomes are being assessed).

2. I would have found it useful to have some sense of why the QMUL Evaluation Toolkit was cited and used. Again, this links back to the framing issue. Does PER really need its own evaluation toolkit given the decades of eg audience research that we can draw on. Is there something unique to PER that justifies a new toolkit? How does the PER Evaluation Toolkit extend or develop existing literature from the fields of eg audience research, or informal science learning? Considering the reference to “informal science learning” I would have expected reference to / critique of the Generic Learning Outcomes framework.

3. The observation that were no examples of long-term impact studies and the suggestion (in line 406) that they “should ideally be done” reinforces that lack of in-depth understanding of the different traditions of evaluation. We know that attending a science festival will be one of many interactions with science (in this case health research) over a lifetime (Archer’s Science Capital work is useful here). These interactions can contradict or reinforce the attitudes to science that the person currently holds when they attend the activity. What that person takes from the experience is affected not only be the activity, but by how they are feeling that day, and what else is going on in their lives outside of attending a festival event. I think it’s naïve to think that it’s possible to track the long-term impacts of a single festival intervention considering the other “noise” that happens in a person’s life before, during, and after participating.

In lines 114-117 you offer a definition of public engagement which seems sensible although I always make a distinction between public engagement with research (PER) and public engagement with a topic (in this case health). PER has to involve the researchers / academics while PE practitioners could do PE with a topic without direct researcher involvement. It would help the paper to be really clear if you are looking at PER or PE with Health. In the section about inclusion criteria (lines 105 and 106) you state that the studies self-identified as being festivals and as being PE. I have seen a lot of things called festivals that weren’t festivals and a lot of things called PE(R) that aren’t PER. Given the small number of examples that were actually looked at, are you confident that they were all festivals and all PER?

I hope these comments are useful and can help to improve the next version – do let me know if anything needs clarifying. I found the methods and results sections very clear to read and understand and I agree with the limitations you identified in the study.

Thank you again for an interesting read.

6. PLOS authors have the option to publish the peer review history of their article (what does this mean?). If published, this will include your full peer review and any attached files.

Reviewer #1: **Yes: **Kofi Aduo-Adjei

Reviewer #2: **Yes: **Helen Featherstone

---

## [Author Response · Author response to Decision Letter 0]

11 Jan 2022

10 January 2022

Dear Editors, 

Re: PONE-D-21-31416 How are public engagement health festivals evaluated? A systematic review with narrative synthesis

Thank you for sending the reviewers’ comments on the above articles. We have revised the paper in light of the comments and respond below on a point-by-point basis. We believe the paper is stronger as a result and would like to thank the reviewers for their helpful feedback. 

We have ensured that the manuscript meets PLOS ONE's style requirements. As this is a review, there is no primary data to make available and reference in the ‘Data Availability Statement’ – all the included papers are referenced in the review together with the search strategy, and the data extracted is presented in a table. We now state in our Data Availability Statement that the article does not contain data and the data availability policy is not applicable. 

We look forward to hearing from you in due course. 

Best wishes, 

Dr Lucy Selman

Reviewer #1

Overview:

This is relevant topic of interest to public health practitioners and researchers in the field of public engagement. Nonetheless, I suggest some substantial reforms to the authors. They are at liberty to address these suggestions to improve the readability and quality of the manuscript.

 Thank you

Topic:

The authors may consider changing the term “systematic review” to “Scoping review”. Mostly systematic reviews are followed with meta-analysis, whiles usually scoping reviews adopt narrative synthesis.

 A scoping review is defined as a type of research synthesis that aims to ‘map the literature on a particular topic or research area and provide an opportunity to identify key concepts; gaps in the research; and types and sources of evidence to inform practice, policymaking, and research’ (Daudt et al. BMC Med Res Methodol 2013). In contrast, narrative synthesis is an approach to the systematic review and synthesis of findings from multiple studies that relies primarily on the use of words and text to summarise and explain the findings of the synthesis (Popay et al. 2006). While scoping reviews may commonly use narrative synthesis, narrative synthesis is also very commonly used in systematic reviews – for just a few recent examples, please see:

Pian et al 2021

Karran et al 2020

Habtewold et al 2020

We have therefore not changed the title or terminology of the methods, as systematic review with narrative synthesis accurately captures our evidence synthesis approach. 

Abstract:

Line 13, author may specifically state the required “findings and recommendation” that may help improve future evaluations

 We have now added the word ‘these’ to the abstract to be clear that we referring to the findings just mentioned:

Higher quality studies had specific evaluation aims, used a wider variety of evaluation methods and had independent evaluation teams. Evaluation sample profiles were often gender-biased and not ethnically representative. Patient involvement in event delivery supported learning and engagement. These findings and recommendations can help improve future evaluations.

Search strategy:

The authors may highlight the justification of searching the listed databases Now added on page 8:

Literature scoping and discussion with a subject librarian helped to inform the choice of databases and the search strategy.

Conclusion:

The authors may highlight certain areas that may require future research concentration based on the review conducted on Public Engagement festivals Thanks for this suggestion. We discuss areas for future research as well as directions for future PE evaluation on page 32, and have added specific wording to clarify this, e.g. 

The results of this review enable us to make some recommendations to evaluators of future health-related PE and suggestions for future research…. With the increasing focus on co-producing and co-delivering health-related PE events with patients and communities, there is a need for future research to understand and assess how the public can best influence festival design, delivery and impact.

Reviewer #2

Thank you for writing this paper and inviting me to comment on it. I found it a clear paper to read and thought the research was interesting, informative and at times surprising.

 Thank you

My criticisms of the paper are not in the research itself, but are in the framing of the paper and lack of real precision about terms used in the paper, which then has implications for the interpretation of the data and conclusions.

The argument for looking at health-related science festivals starts from a public engagement with research (PER) frame, with festivals being a potential site of PER, and health-related science festivals being a subsection of festivals and being worthy of investigation. I am not convinced that this is the most robust case for looking at health-related science festivals. I was surprised, for example, that there was no mention of the long-standing practice of PPI in health research and how PER at festivals relates to PPI. An alternative argument would be to start with science festivals in general, how they can be a site for PER, and for health PER specifically. There are others!

 Thank you for your really helpful engagement with the review and your suggestions to clarify our use of terms and revisit how we frame the review. 

We have now added a substantial discussion of public engagement and patient and public involvement (definitions and differences) to the Introduction, referencing the importance of both within health research: 

There are many overlaps between public engagement (PE) and the long-standing practice of Patient & Public Involvement (PPI) in medical and healthcare research. Commonly accepted definitions of these terms are given below, as set out by leading organisations in these two fields, the National Coordinating Centre for Public Engagement (NCCPE) and the National Institute for Health Research (NIHR) respectively.

Public Engagement: The myriad of ways in which the activity and benefits of higher education and research can be shared with the public. Engagement is by definition a two-way process, involving interaction and listening, with the goal of generating mutual benefit.[5] 

Patient & Public Involvement: Research being carried out ‘with’ or ‘by’ members of the public rather than ‘to’, ‘about’ or ‘for’ them. It is an active partnership between patients, carers and members of the public with researchers that influences and shapes research.[6]

The definitions above demonstrate that whilst PPI is a relatively tightly defined concept as understood by healthcare practitioners and researchers, PE is a much more amorphous term [7,8], encompassing many ways of engaging with the public and not necessarily just about research. PE, particularly when it involves engagement with specific research projects, or research-related matters (e.g. research ethics), rather than engagement around a wider subject area or topic, is sometimes specifically referred to as ‘Public Engagement with Research’ (PER). This part of the engagement spectrum is where there are most overlaps with PPI. Whereas PPI is generally a formally defined process within a healthcare research project, PE activities are often more informal, sometimes ad-hoc and can be delivered in a multitude of ways for a wide variety of audiences.

If the precise meaning of engagement is vague, then an catch-all definition of ‘the public’ is even harder to pin down [11, 12]. What is commonly accepted in the PE sphere is that ‘the public’ should never be considered as one single entity, but a multi-dimensional spectrum of people with widely varying levels of expertise, lived experiences, interests, opinions and so on [13, 14]. It is critical that any PE activity is tailored to the specific audience it is aimed at, perhaps even co-developed with that group of people. In the context of this paper the understanding of ‘publics’ as “gatherings of people, things, objects and ideas convened around a matter of concern.” as derived by Facer (2020) [15] is helpful.

PPI and PE both play an important role in research related to human health. The UK’s National Health Service (NHS) and the USA’s Institute of Medicine support the co-production of healthcare plans with patients, increasing patient control over their health and emphasising disease prevention [16, 17]. People may therefore, more than ever, have reason to seek out and engage with health-related research. While relatively few members of the public have the opportunity to take part in a formal PPI process, public engagement opportunities might more readily present themselves. 

We go on to discuss science festivals as one increasingly popular format for communication of, and public engagement with, health research. 

In the paper there are mentions of “audience experience”, “informal science learning”, “visitor studies”, “health literacy”, alongside “public engagement with research” but there is little demonstration of an understanding of the differences and similarities in evaluation that arise from these different frames. An understanding of these different areas of practice would have helped with interpreting the data. In particular there were three areas where this came through: how publics are presented, why the QMUL toolkit was used, and the observation of there being no longitudinal work presented:

1. In the paper we encounter terms such as “general public”, “lay citizen”, “patients”, “users of health services”, “professionals”, “citizens” and “audiences”. There is a substantial body of work that examines what we mean when we use these terms, for example we would largely argue that there is no such thing as a “general public”. We can describe how publics form or respond to particular interventions or circumstances. We can think about what the publics’ roles are in the engagement experience: are they audiences there to listen? participants with experience / insight to share? We can think about current levels of knowledge / attitude / behaviour with respect to the topic under discussion. We can refute the idea that all publics are citizens. This lack of critical engagement with these concepts, particularly in the different areas of evaluation practice mentioned in the previous paragraph, comes through repeatedly in the introduction and therefore in the interpretation of the data (particularly with respect to what outcomes are being assessed).

 We agree that different approaches to evaluation are reflected in and associated with the use of different kinds of terms in public engagement. However, it was not in the remit of this this review to examine differences in evaluation approaches associated with different PE frames or traditions. Our aims were more practical than theoretical in this respect. In response to your specific points: 

1. We have now added to the Introduction a passage reflecting on definitions of the public which includes the following: 

What is commonly accepted in the PE sphere is that ‘the public’ should never be considered as one single entity, but a multi-dimensional spectrum of people with widely varying levels of expertise, lived experiences, interests, opinions and so on [13, 14]. It is critical that any PE activity is tailored to the specific audience it is aimed at, perhaps even co-developed with that group of people. In the context of this paper the understanding of ‘publics’ as “gatherings of people, things, objects and ideas convened around a matter of concern.” as derived by Facer (2020) is helpful [15].

In our interpretation of the data, we use the evaluation outputs and outcomes defined by Grant (2011). We have added the following point to the Discussion to reflect the comment regarding how concepts and frameworks link to evaluation: 

Related to this, evaluations would benefit from more explicit discussion of the aims, framework and assumptions underlying PE initiatives. Differences in conceptualisation and fundamental approach (e.g. regarding the role of the public in the engagement experience) have implications for the choice of appropriate outcomes and evaluation methodology [8, 49, 50].

2. I would have found it useful to have some sense of why the QMUL Evaluation Toolkit was cited and used. Again, this links back to the framing issue. Does PER really need its own evaluation toolkit given the decades of eg audience research that we can draw on. Is there something unique to PER that justifies a new toolkit? How does the PER Evaluation Toolkit extend or develop existing literature from the fields of e.g. audience research, or informal science learning? Considering the reference to “informal science learning” I would have expected reference to / critique of the Generic Learning Outcomes framework.

 2. Our aim in this review was to synthesise evidence on health-related PE festivals from a range of traditions/using different framing of concepts. One of the identified studies investigated ‘informal science learning’, however this type of PE was not a focus of the review. The QMUL toolkit seemed appropriate as a comparator in this review given the different approaches and frameworks employed in the studies (most of which were not explicit about which PE research/tradition they were based on). The toolkit is widely used by health researchers and has been developed in an attempt to strengthen the evaluation of academic PE initiatives and simplify the process. Many non-PE academics are unfamiliar with the theories and traditions of PE. However, rather than explicate these our aim was to add to the utility of our review by comparing published evaluations with a well-recognised toolkit. We have no connection with the toolkit or its development/use, in case that was a concern here. 

We have added the following on page 7:

The Queen Mary University of London (QMUL) public engagement evaluation toolkit [25, 26] has been developed as an open-access, pragmatic, generic toolkit applicable to diverse forms of academic PE and proposed as a “common ‘evaluation standard’” [27]. 

We chose the QMUL toolkit as an appropriate comparator for this review as it is familiar to health researchers [29] and is applicable to a wide variety of engagement activities, such as the studies included in this review, which utilise multiple different approaches and frameworks.

3. The observation that were no examples of long-term impact studies and the suggestion (in line 406) that they “should ideally be done” reinforces that lack of in-depth understanding of the different traditions of evaluation. We know that attending a science festival will be one of many interactions with science (in this case health research) over a lifetime (Archer’s Science Capital work is useful here). These interactions can contradict or reinforce the attitudes to science that the person currently holds when they attend the activity. What that person takes from the experience is affected not only be the activity, but by how they are feeling that day, and what else is going on in their lives outside of attending a festival event. I think it’s naïve to think that it’s possible to track the long-term impacts of a single festival intervention considering the other “noise” that happens in a person’s life before, during, and after participating.

 Thanks for these interesting points. Our intention was not to imply that long-term follow up was needed in the sense of identifying the impact of a specific festival in someone’s life separate to the ‘noise’ of other factors. Rather, we were referring to the fact that the evaluations we identified focus on immediate impact (e.g. views immediately after the festival), i.e. outcomes which are extremely proximal to the event. There is little attempt to ask for more reflective opinions or ascertain whether e.g. changes in behaviour posited as a result of the festival were actual rather than potential. For example, someone could report that at a festival they learnt about how antibiotics function and they thought this would change their behaviour, and it would be relevant to ask at a later timepoint whether this learning had actually resulted in changes in behaviour or information sharing with others. We believe these kinds of outcomes would be relevant and interesting – not to ‘track’ long-term effects as such, but to unpick the complex effects and influences at play (which much evaluation of other interventions tries to do, e.g. through the development of logic models or using a realist approach). We have modified the relevant section to reflect this (p.30): 

Assessing the longer term and broader impacts of festival activities can be practically difficult within a time-limited research grant, but more reflective opinions of a festival and accounts of whether, for example, potential changes in behaviour translated into actual changes in behaviour, might still be relevant, especially when PE festivals are ongoing [18,21,27,51]. Such longer-term evaluations could help explain the complex effects and interactions at play and help develop a better understanding of active ingredients and mechanisms of action in PE via festivals. 

In lines 114-117 you offer a definition of public engagement which seems sensible although I always make a distinction between public engagement with research (PER) and public engagement with a topic (in this case health). PER has to involve the researchers / academics while PE practitioners could do PE with a topic without direct researcher involvement. It would help the paper to be really clear if you are looking at PER or PE with Health. Thanks for flagging this – as it wasn’t a requirement for researchers or academics to be present at the festival, we have edited the definition to be clear about this (p.7): 

Two-way dialogue between health-related researchers (including social scientists) or PE practitioners and members of the general public [3]. We focus here on engagement in relation to health-related research or a health topic, including medicine and applied health. 

In the section about inclusion criteria (lines 105 and 106) you state that the studies self-identified as being festivals and as being PE. I have seen a lot of things called festivals that weren’t festivals and a lot of things called PE(R) that aren’t PER. Given the small number of examples that were actually looked at, are you confident that they were all festivals and all PER?

 We chose to focus on studies which self-identified as evaluations of festivals, and used our definition of ‘festival’ to determine whether or not their self-description should be included in the review. We could have chosen another approach and reviewed all health PE evaluations to identify those which met the definition, however that much wider review was not possible within the resource constraints of this project. It would, though, be an interesting approach in a future evidence synthesis for the reasons you have outlined. We have added to our Limitations (p.31): 

Limitations related to the resources available for the review include not searching the grey literature, using only adult data, restricting the review to English language study reports only, and requiring that authors self-identified their PE as a festival. 

There is always some degree of subjectivity involved in applying inclusion and exclusion criteria, however we attempted to minimise this by: scoping and considering the literature as a first step, developing precise inclusion/exclusion criteria based on established definitions of PE and ‘festival’, and considering complex cases as a team to determine the application of the criteria. We are confident that the included studies meet the inclusion criteria.

I hope these comments are useful and can help to improve the next version – do let me know if anything needs clarifying. I found the methods and results sections very clear to read and understand and I agree with the limitations you identified in the study.

Thank you again for an interesting read.

 Thank you for your very helpful comments.

---

## [Decision Letter · Decision Letter 1]

4 Apr 2022

How are public engagement health festivals evaluated? A systematic review with narrative synthesis

PONE-D-21-31416R1

Dear Dr. Selman,

We’re pleased to inform you that your manuscript has been judged scientifically suitable for publication and will be formally accepted for publication once it meets all outstanding technical requirements.

Kind regards,

Professor Benjamin Tan, BNSc MMed PhD RN

Additional Editor Comments (optional):

The reviewers' comments have been well addressed in the revised manuscript. The academic editor is satisfied with your revision. A decision has therefore been made to accept your paper for publication.

Reviewers' comments:

Reviewer's Responses to Questions

**Comments to the Author**

1. If the authors have adequately addressed your comments raised in a previous round of review and you feel that this manuscript is now acceptable for publication, you may indicate that here to bypass the “Comments to the Author” section, enter your conflict of interest statement in the “Confidential to Editor” section, and submit your "Accept" recommendation.

Reviewer #1: All comments have been addressed

Reviewer #2: All comments have been addressed

2. Is the manuscript technically sound, and do the data support the conclusions?

Reviewer #1: Yes

Reviewer #2: Yes

3. Has the statistical analysis been performed appropriately and rigorously? 

Reviewer #1: Yes

Reviewer #2: N/A

4. Have the authors made all data underlying the findings in their manuscript fully available?

Reviewer #1: Yes

Reviewer #2: Yes

5. Is the manuscript presented in an intelligible fashion and written in standard English?

Reviewer #1: No

Reviewer #2: Yes

6. Review Comments to the Author

Reviewer #1: (No Response)

Reviewer #2: (No Response)

7. PLOS authors have the option to publish the peer review history of their article (what does this mean?). If published, this will include your full peer review and any attached files.

Reviewer #1: No

Reviewer #2: **Yes: **Helen Featherstone, PhD

---

## [Editor Report · Acceptance letter]

12 Aug 2022

PONE-D-21-31416R1 

How are public engagement health festivals evaluated? A systematic review with narrative synthesis 

Dear Dr. Selman:

I'm pleased to inform you that your manuscript has been deemed suitable for publication in PLOS ONE. Congratulations! Your manuscript is now with our production department. 

Kind regards, 

on behalf of

Professor Benjamin Tan 

Academic Editor

PLOS ONE